# Highly Water-Repellent and Anti-Reflective Glass Based on a Hierarchical Nanoporous Layer

**Shuntaro Minegishi, Nanako Ueda, Mizuki Saito, Junhwan Lee** and **Takuya Fujima** *

Faculty of Engineering, Tokyo City University, 1-28-1 Tamazutsumi, Setagaya, Tokyo 156-8557, Japan; g2081052@gmail.com (S.M.); uedananako7@gmail.com (N.U.); g2181022@tcu.ac.jp (M.S.); g2181056@tcu.ac.jp (J.L.)
* Correspondence: tfujima@tcu.ac.jp

**Abstract:** Optically anti-reflective and water-repellent glass is required for solar cell covers to improve power-generation efficiency due to transparency improvement and dirt removal. Research has been conducted in recent years on technologies that do not use fluorine materials. In this study, we focused on the anti-reflective properties and microstructure of hierarchical nanoporous layer (HNL) glass and used it as a substrate. As a result, we have achieved both strong anti-reflectivity and high water repellency on HNL glass by coating polydimethylsiloxane (PDMS) using baking and thermal chemical vapor deposition (CVD). The surfaces showed a significantly higher sliding velocity of water droplets than the PDMS-treated material on the flat glass plate. They also showed such water repellency that the droplets bounced off the surface.

**Keywords:** anti-reflection; hydrophobicity; water repellency; hierarchically porous structure

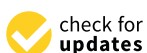



## 1. Introduction

Renewable energy development is a global issue, and various power generation technologies are actively being developed. Photovoltaic power generation has reached a practical level of power generation efficiency after decades of research and is currently the largest renewable energy source. However, studies have shown that the power generation efficiency of photovoltaic modules decreases by as much as 50% yearly due to the adhesion of dust to the surface [1,2]. Additionally, light reflection on the cover glass surface also causes losses in energy efficiency [3]. The prevention of dust adhesion by hydrophobic coating and the reduction in reflectance due to anti-reflective coating are often studied to reduce these energy losses [2]. Superhydrophilic surfaces are also thought to have anti-fouling properties due to a different mechanism: strong hydro-affinity makes water penetrate between the surface and attached fouling to make it flow away. However, the combination of long-lasting superhydrophilicity and anti-reflectivity is quite rare [4].

Hydrophobic coatings are often produced by depositing hydrophobic materials on the surface of a substrate by painting, baking, or vapor deposition. Fluorinated and silicone polymers have commonly been used as hydrophobic agents. Still, fluorinated hydrophobic agents are toxic to human health, and their use has been restricted internationally since the Stockholm Convention (2009) [5–7]. Therefore, there is a need to use coatings with silicone hydrophobic agents and to improve their functionality.

PDMS is a silicone hydrophobic agent with Si-O as its main chain. Due to its non-toxicity, PDMS is expected to be an alternative material to fluorinated hydrophobic agents and has been extensively researched [8,9]. Although a flat PDMS surface exhibits a water-contact angle of ca. 90°, nano- to micro-structures have increased the hydrophobicity with water-contact angles of over 150°, maintaining the glass substrate's transparency [10–13].

There are two methods to form anti-reflection (AR) coatings. The first is stacking thin dielectric layers with different refractive indices to control the reflection by interference. The other is developing a fine structure on the material surface to change the effective

refractive index gently, as occurs with the structure of a moth eye [14–17]. The former method uses a vapor deposition method, which is expensive to process, and the range of light incidence angles over which the effect can be obtained is limited. The latter method has a wide range of effective incident angles, but its uneven structure poses problems in terms of strength and dirt removal. Min Wang et al. achieved a low reflectance of about 5% and a superhydrophobicity of about a 155° water droplet-contact angle using micron-sized irregularities, but they used PTFE [15]. Xiaoyu Sun et al. used a sol-gel method to coat PTFE, resulting in a porous film with a maximum visible light transmittance of 97.86%, albeit non-uniform within 400–800 nm, and hydrophobicity of approximately 100° at the water droplet contact angle [17].

Hierarchical nanoporous layer (HNL) glass is a functional material in which a porous layer is formed on the surface of silicate glass by a simple one-step process. It has a pore size of several tens of nanometers at the apparent surface, which gradually decreases toward the depth direction. The structure can realize optical anti-reflective properties of less than 0.5% in a wide wavelength range and superhydrophilic properties that maintain a water-contact angle of less than 5° for a long time [4].

This study attempts to fabricate a functional glass with water removal and anti-reflection properties by coating PDMS on HNL glass using thermal CVD and baking. This glass can improve the efficiency of photovoltaic power generation at a low cost because the fabrication method is easier than the sol-gel method and the water-repellent agent is much less expensive than toxic fluorine agents. In addition, the ability to fabricate porous structures on large glass is attractive for future mass production. We evaluated static wettability using a water-contact angle and dynamic wettability by a water-sliding angle and sliding velocity [18,19].

## 2. Experimental Section

We used borosilicate glass (TEMPAX Float®; Schott Jenaer Glas GmbH, Jena, Germany) as the substrate and PDMS (KF96-50cs; Shin-Etsu Chemical Corp., Tokyo, Japan) as the water-repellent agent. We formed HNL on the glass according to the reference [4]; the HNL was formed by heating the pristine glass in a sodium bicarbonate solution at 110 °C for 28 h. The pristine glass was also used for comparison.

We performed thermal CVD on substrates by heating a vessel to 300 °C, in which substrates were suspended, with PDMS on the bottom. We also spin-coated PDMS onto the substrates and then baked it at 300 °C. The heating duration varied, ranging from 20 to 420 min for both methods. The obtained samples were evaluated after ultrasonication in toluene and purified water.

The contact angle of a drop of water was determined by the θ/2 method using a contact angle meter (PCA-1; Kyowa Interface Science Corp., Saitama, Japan). The water droplets used were four µL of purified water, based on ISO 19403.

The water-sliding angle was defined as the angle at which both the front and rear endpoints of the water droplet began to move when the sample was tilted after a water drop was put on the horizontal sample surface. Purified water samples measuring 20, 30, and 40 µL were used for the water droplets. The displacement of the front endpoint of the droplet was measured using a video to evaluate the removal velocity.

We also demonstrated and observed a water droplet dropping on the samples using a high-speed camera to observe a 10 µL drop of purified water from a height of 3 cm.

We evaluated the optical reflectivity of the samples using a microscopic single-sided reflectivity spectrometer (XSP-100; Shibuya Optical Corp., Saitama, Japan). The spectra covered a wavelength range of 400 to 900 nm. We also observed the surface and cross section of the samples using scanning electron microscopy (FE-SEM SU-8230; Hitachi High-Tech Corp., Tokyo, Japan).

### 3. Results and Discussion

PDMS treatments significantly decreased static wettability: the water contact angles of the HNL and pristine glass were less than 5° and about 13° before PDMS treatments but about 140° and 90° after the treatments, respectively. As shown in Figure 1, the water contact angles were independent of the two treatment methods and the treatment time. The contact angle for the PDMS-treated pristine glass is consistent with reported angles for PDMS. Still, HNL is thought to have increased its contact angle due to the Cassie–Baxter effect on the rough apparent surface at the end of its porous structure.

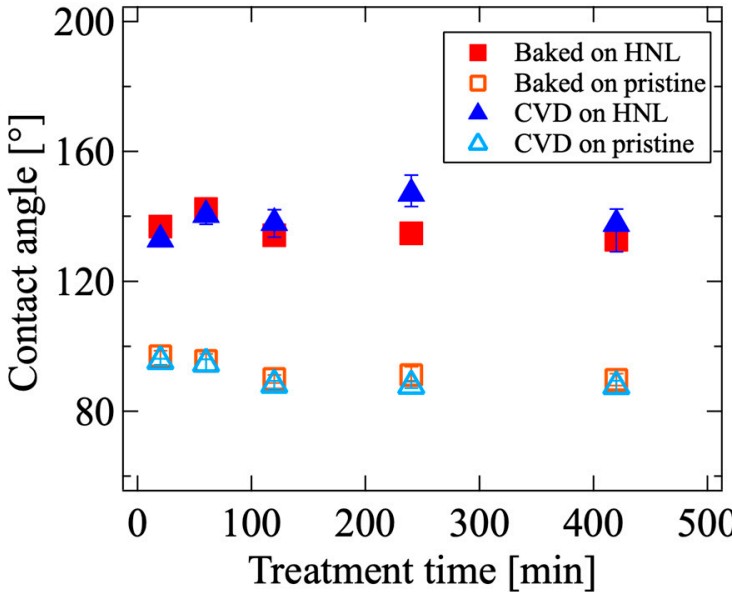

**Figure 1.** Water-contact angles on PDMS-coated HNL and pristine glass. HNL-based samples exhibited higher hydrophobicity for both baking and CVD methods.

Figure 2 shows the sliding angle of samples prepared by PDMS baking and the thermal CVD. The figure shows that the sliding angle for both substrates changed with processing time following a similar trend, with the highest water repellency at 120 min. On the other hand, the trend in the sliding angle for the thermal CVD samples was different for HNL and pristine glass. The water droplets did not slide on the HNL sample of 20 min treatment, and a significant deterioration in water repellency occurred after 420 min of treatment, while the pristine sample showed the same trend as the baked samples.

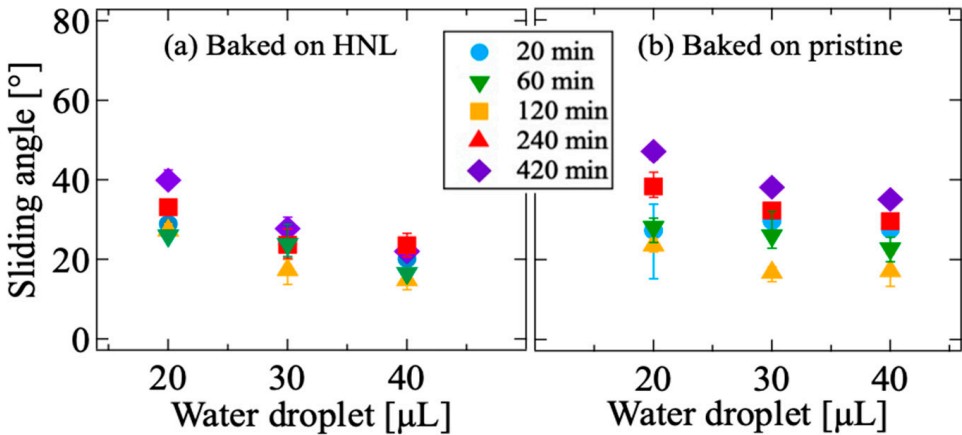

**Figure 2.** *Cont.*

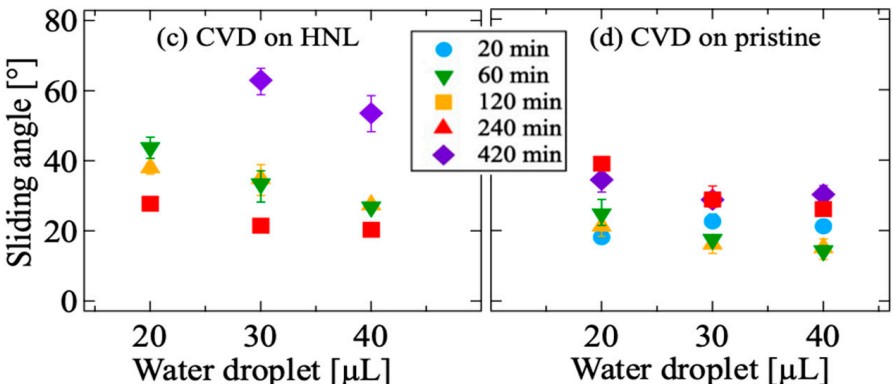

**Figure 2.** Sliding angles of water droplets on the PDMS-coated glass prepared (**a**,**b**) by baking and (**c**,**d**) by thermal CVD. CVD took more time to make HNL hydrophobic than baking, though the methods worked similarly on pristine glass.

The water sliding velocity at a tilt angle of 20° was evaluated for samples with the same treatment time, i.e., 120 min (Figure 3). Compared to the pristine substrate samples, the HNL substrate sample shows a significantly higher water-repellent velocity for both CVD and baking treatments.

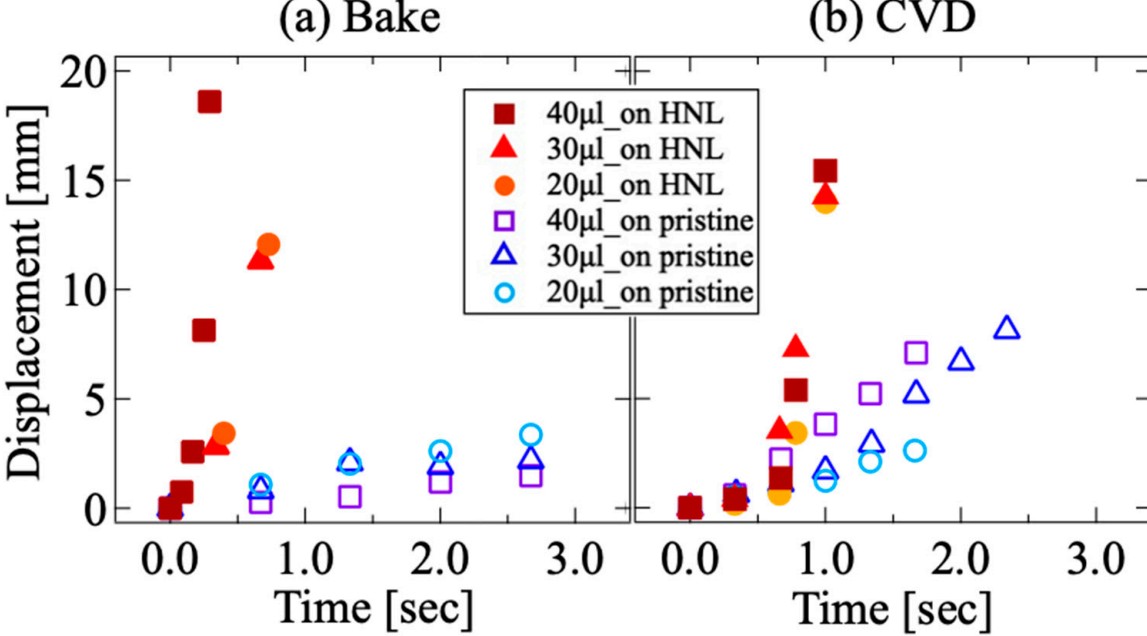

**Figure 3.** The displacement of water droplets on samples PDMS-coated (**a**) by baking and (**b**) by CVD. The HNL-substrate samples exhibited significantly large sliding velocity for both coating methods.

This difference in water repellency is also apparent in more dynamic phenomena. For example, Figure 4 shows the behavior of water droplets as they fall on the sample surface treated for 120 min, as captured by a high-speed camera. This work realized the water-bouncing hydrophobic surface without any fluorine agents other than similar surfaces developed so far [20,21].

The water droplet was deformed with momentum on the PDMS-baked sample with the pristine substrate but remained on the surface after hitting the sample surface (Figure 4a). This behavior was similar to that observed for CVD-treated samples on the pristine substrates.

## a) Baked on pristine

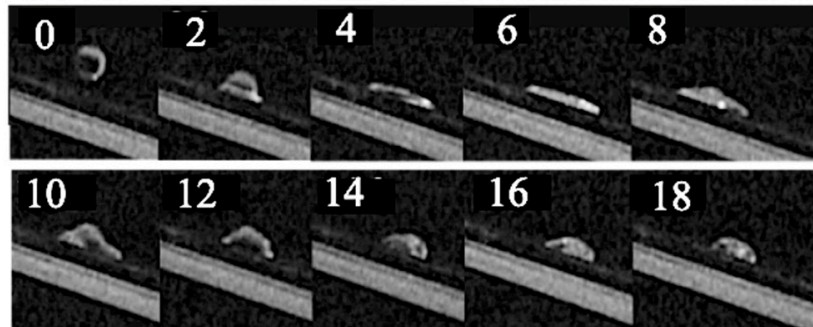

## b) Baked on HNL

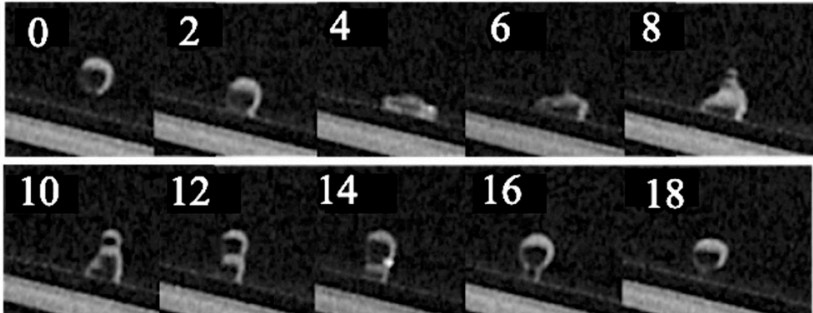

## c) CVD on HNL

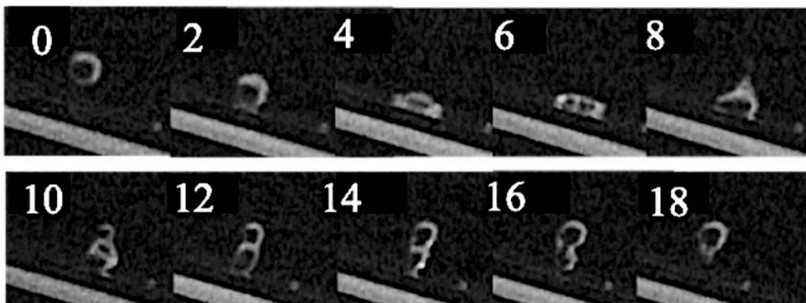

**Figure 4.** High-speed camera view of a water drop falling on samples (**a**) baked on the pristine substrate, (**b**) baked on the HNL, and (**c**) by CVD on HNL. The number in each frame indicates the time (in ms) elapsed from the moment of the drop.

On the other hand, the water droplets were similarly deformed and then moved away from the surface with the momentum of the recovery of their shape due to surface tension on the PDMS-baked sample with the HNL substrate (Figure 4b). In other words, the water droplets bounced on the sample, suggesting that the surface is highly water-repellent. The water droplets jumped even higher on the CVD-treated sample with the HNL substrate (Figure 4c), indicating an even higher water repellency.

The reflectivity spectra of the PDMS-treated HNL surfaces are shown in Figure 5. Both baking and CVD samples maintained significant AR properties compared to pristine glass, with the effect of thin-film interference changing slightly with treatment time. In other words, PDMS-treated HNL glass exhibited both water repellency and low reflectivity at a high level.

Figure 6 shows SEM micrographs of the sample surface with the HNL substrate. PDMS fills the open pores on the HNL surface with time in the baking process, but this does not occur in the CVD process, according to Figure 6a,c. However, since the vacancies remain

even after baking, the Cassie–Baxter state occurred for both methods. It resulted in the high hydrophobicity shown in Figure 1 and the water repellency shown in Figures 3 and 4.

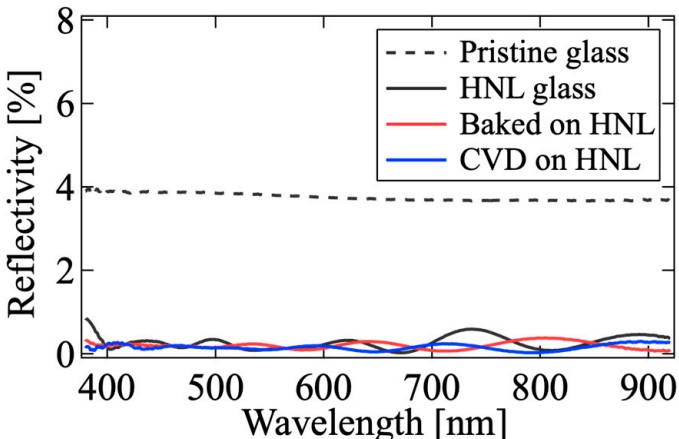

**Figure 5.** The reflectivity spectra of the PDMS-treated (120 min) HNL glass compared to the pristine and untreated HNL glass. Both samples treated by baking and CVD maintained the anti-reflection property that untreated HNL intrinsically has.

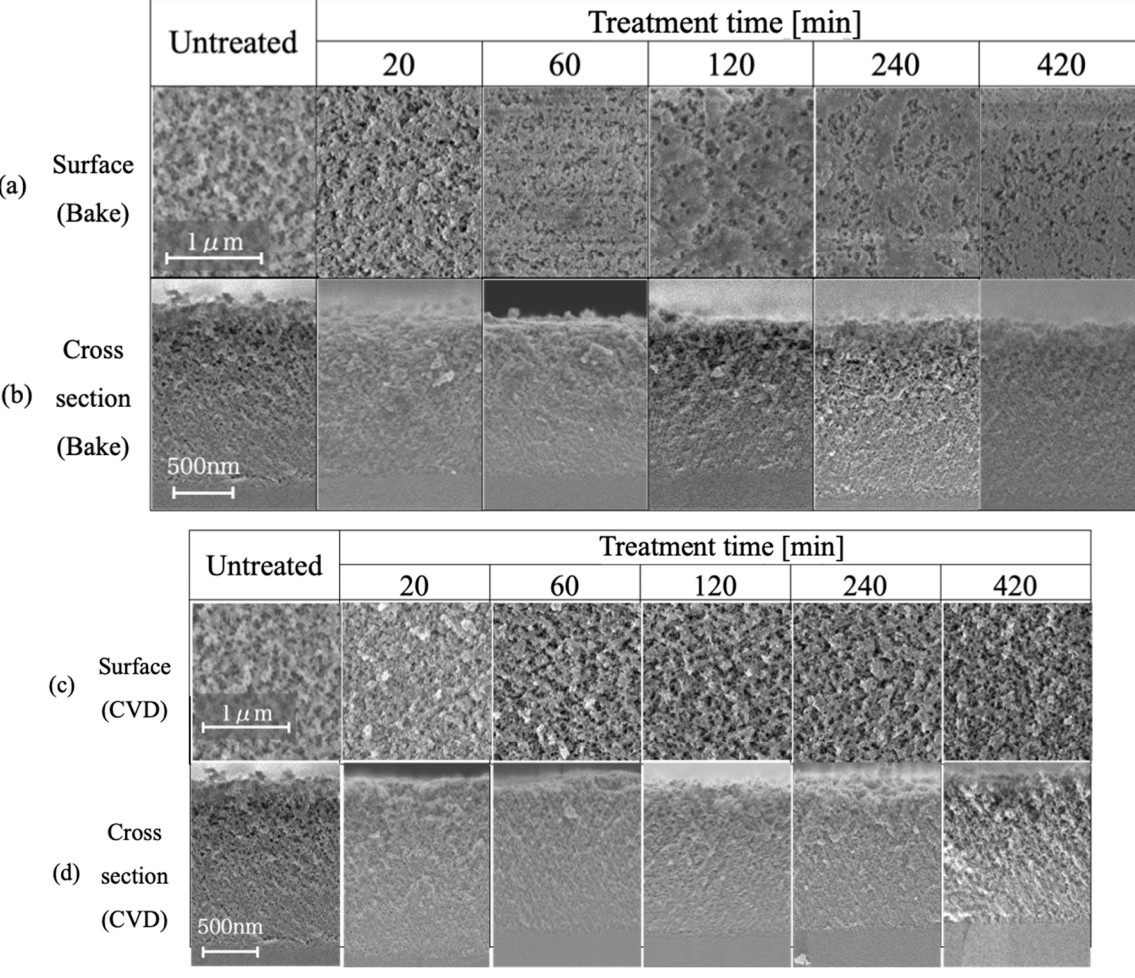

**Figure 6.** SEM micrographs of (**a**), (**b**) the PDMS-baked HNL glass and (**c**), (**d**) the CVD-treated HNL glass. The surface view (**a**,**c**) and cross-sectional view (**b**,**d**) indicate that the HNL was not filled by PDMS other than most of the surface area.

Figure 6b,d show that the pores in the HNL layer were still open even after long treatments and preserved the unique structure of the HNL. This also maintained the gradual decrease in the effective refractive index from the surface to the bulk, indicating that the AR property was maintained.

Here, we can find a correlation between the water repellency and the SEM micrographs of the HNL substrate samples. Water droplets showed larger acceleration on the baked sample in Figure 3, and water droplets bounced higher on the CVD sample in Figure 4. These differences in water repellency should come from the microscopic shape of the sample surface. The baked samples had a relatively flat surface that moved water droplets quickly in the in-plane direction [22]. On the other hand, the CVD sample, keeping the HNL shape almost intact, was not as smooth as the baked sample but more easily detached water in the out-of-plane direction.

Table 1 compares the material obtained in this work with other highly water-repellent and anti-reflective surfaces with [14] and without fluorine agents [23–25]. Although the materials in this work are not lower than those of other works in terms of the sliding angle, they feature powerful anti-reflective effects and simple and inexpensive materials and fabrication methods. These features are of great advantage for practical large-scale applications and make the materials promising candidates for solar panel covers with excellent water repellency, water droplet removal performance, and visible light transmittance.

**Table 1.** Specification and performance comparison of water-repellent and anti-reflective surfaces.

| | This Work | A. B. Gurav, et al. [23] | W Dong, et al. [24] | X Liu, et al. [25] | M Wang, et al. [14] |
|---|---|---|---|---|---|
| **Substrate** | Borosilicate glass | Glass slide | Glass slide | Glass slide | Polycrystalline silicon solar cell |
| **Agent** | PDMS | TEOS/EtOH/NH$_4$OH/HMDZ, Silicone-oil | Methyltrimethoxysilane, Isopropanol | PDMS/TEOS/DBTDL /n-hexane | PTFE |
| **Method** | Baking/Thermal CVD | Sol-gel SiO$_2$ coating | Coating by wiping, followed by heating | calcining candle-soot-coated | A facile hot embossing lithography process |
| **Contact angle** | 140° | 166° | 88° | 163° | 155° |
| **Sliding angle** | 20° | <4° | 17° | <1° | 15° |
| **Visible light transmittance** | Reflectivity <0.5% | 91%~98% | 96.7% | 89.5% | Reflectance 5% |

## 4. Conclusions

We tried to realize a glass that simultaneously offers optical anti-reflectivity and high water repellency without fluorine agents. We accomplished this with a PDMS coating on HNL glass, which exhibits long-lasting superhydrophilicity and anti-reflectivity over a wide range of wavelengths. Two types of samples created by different coating methods, baking and thermal CVD, showed comparable water-contact angles and water-sliding angles but different microstructures on their surfaces. The baked sample, in which the open pores remained but were partially filled and close to a smooth surface, had a faster water sliding speed and better in-plane directional water repellency. On the other hand, the CVD-treated sample, which left the porous structure of HNL intact, had a better bounce back when water droplets hit it and was superior in removing water droplets in the out-of-plane direction. These different treatment methods should be used while considering the installation environment, and they are expected to have practical applications in products such as solar panel covers, where both anti-reflectivity and water repellency are required. Hence, a substantiative experiment on the solar panel cover should be conducted for the next step. While it is technically easy to bake or vapor-deposit PDMS on large-format glass, HNL formation requires further development of manufacturing equipment.

**Author Contributions:** Conceptualization and methodology, S.M. and T.F.; resources, S.M., M.S. and J.L.; formal analysis and investigation, S.M.; validation, S.M. and N.U.; data curation and writing—original draft preparation, S.M.; writing—review and editing, T.F.; visualization, S.M.; supervision, project administration, and funding acquisition, T.F. All authors have read and agreed to the published version of the manuscript.

**Funding:** This research was funded by Japan Science and Technology Agency, A-STEP JPMJTR20TK.

**Institutional Review Board Statement:** Not applicable.

**Informed Consent Statement:** Not applicable.

**Data Availability Statement:** The authors confirm that the data supporting the findings of this study are available within the article.

**Acknowledgments:** The authors appreciate T. Kuroiwa (Tokyo City University) for fruitful discussion.

**Conflicts of Interest:** The authors declare no conflict of interest.

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
