# Peer review of "Highly Water-Repellent and Anti-Reflective Glass Based on a Hierarchical Nanoporous Layer"

_coatings, doi:10.3390/coatings12070961_

Round 1

Reviewer 1 Report

Dear Authors

The topic of your manuscript is interesting and well elaborated. The manuscript is well organized, however it needs improvement in terms of a novel set of references (most of them are from older date). I hope that you can find newer ones in order to accompany your study.

Some minor suggestions are given in the attached copy. I hope that it will improve your article.

Thank you.

Author Response

Dear Referee-1

   The authors appreciate referee-A's comments and have added new references for discussion. Other suggestions provided in the PDF also helped the authors to improve the manuscript, especially the mention of hydrophilic surfaces.

Thank you again for your advice.

Reviewer 2 Report

The authors present a hierarchical nano-porous coating for water repellent and and anti reflective glass solar cells. The research is clear and concise and the results are interesting. However,  there are some technical and analytical errors that need attention for publication. Hence, I would like to decide based on the major revision of the following comments.

1- The manuscript title is grammatically and structurally incorrect. It is an incomplete sentence. Follow MDPI instructions to authors to write a perfect title and revise the title for better understanding.

2- There are plenty of recent studies on the research topic with strong results. Clearly explain the novelty of the study that distinguishes the results from the recently published papers?

3- I recommend the authors to provide a comparative analysis of the results in the present work with the recently published papers in a tabular form. If true, explain the reason for the improved results in the main body of the manuscript.

4-  Provided the fact that the coatings are used on a large scale windows on glass for green environment, what is the economic value of the nanoporous coatings for large scale development?

5- The results are descriptive rather argumentative. It is highly recommended for the authors to present the results in argumentative manner with proper references rather than a simplistic description of the figures.

6- Provide the future directions and the limitations to this study in the conclusion section.

Author Response

Dear Referee-2

   The authors appreciate referee-A's comments and advice to improve the manuscript. Here, the point-by-point replies to the reviewer's comments follow. 

1) The authors have changed the title for a better understanding of the readers.

2) The authors have added recent studies as references and made it more apparent that the advantage of the surface developed in this work is the combination of water repellency and anti-reflectivity "without fluorine agent."

3) Accepting the reviewer's recommendation, the authors have added a comparison table at the end of the "results and discussion" part.

4) The mass production effect on the cost is hard to estimate at this moment, partially because the large-scale HNL manufacturing equipment is still under development. Therefore, the authors have added a discussion about the economic value only based on the material and process cost.

5) The authors modified the "results and discussion" part to be more argumentative. The comparison table recommended in comment 6 was also helpful.

6) According to the reviewer's advice, the authors have added comments about the future directions and the limitations in the conclusion section.

The authors hope the revised manuscript is suitable for publication in coatings.

Round 2

Reviewer 2 Report

The revised manuscript looks in good shape and is acceptable for publication.

Good Luck